# Role of Brachytherapy in the Postoperative Management of Endometrial Cancer: Decision-Making Analysis among Experienced European Radiation Oncologists

**DOI:** 10.3390/cancers14040906

**Published:** 2022-02-11

**Authors:** Markus Glatzer, Kari Tanderup, Angeles Rovirosa, Lars Fokdal, Claudia Ordeanu, Luca Tagliaferri, Cyrus Chargari, Vratislav Strnad, Johannes Athanasios Dimopoulos, Barbara Šegedin, Rachel Cooper, Esten Søndrol Nakken, Primoz Petric, Elzbieta van der Steen-Banasik, Kristina Lössl, Ina M. Jürgenliemk-Schulz, Peter Niehoff, Ruth S. Hermansson, Remi A. Nout, Paul Martin Putora, Ludwig Plasswilm, Nikolaos Tselis

**Affiliations:** 1Department of Radiation Oncology, Kantonsspital St. Gallen, 9007 St. Gallen, Switzerland; paulmartin.putora@kssg.ch (P.M.P.); ludwig.plasswilm@kssg.ch (L.P.); 2Department of Oncology, Aarhus University Hospital, 8200 Aarhus, Denmark; karitand@rm.dk (K.T.); Lars.Fokdal@auh.rm.dk (L.F.); 3Department of Radiation Oncology, Hospital Clinic Barcelona, 08036 Barcelona, Spain; rovirosa@ub.edu; 4Fonaments Clinics Department, Faculty of Medicine, Universitat de Barcelona, 08036 Barcelona, Spain; 5Department of Radiotherapy, Institute of Oncology “Prof. Dr. Ion Chiricuta”, 400015 Cluj-Napoca, Romania; claudia_ordeanu@yahoo.com; 6Dipartimento di Diagnostica per Immagini, Radioterapia Oncologica ed Ematologia, Fondazione Policlinico Universitario “A. Gemelli” IRCCS, 00168 Roma, Italy; luca.tagliaferri@policlinicogemelli.it; 7Department of Radiation Oncology, Gustave Roussy Comprehensive Cancer Center, 94805 Paris, France; cyrus.chargari@gustaveroussy.fr; 8Department of Radiation Oncology, University Hospital Erlangen, 91054 Erlangen, Germany; vratislav.strnad@uk-erlangen.de; 9Department of Clinical Therapeutics, School of Medicine, National and Kapodistrian University of Athens, 11528 Athens, Greece; adimopoulos@metropolitan-hospital.gr; 10Department of Radiation Oncology, Institute of Oncology Ljubljana, Faculty of Medicine, University of Ljubljana, 1000 Ljubljana, Slovenia; bsegedin@onko-i.si; 11Radiotherapy Research Group, Leeds Cancer Centre, St. Jame’s University Hospital, Leeds LS9 7TF, UK; rachel.cooper1@nhs.net; 12Department of Oncology, Oslo University Hospital, 0424 Oslo, Norway; uxnakk@ous-hf.no; 13Department of Radiation Oncology, University Hospital Zurich, 8091 Zurich, Switzerland; primoz.petric@usz.ch; 14Department of Radiotherapy, Radiotherapiegroep Arnhem, 6815 AD Arnhem, The Netherlands; e.vandersteen-banasik@radiotherapiegroep.nl; 15Department of Radiation Oncology, Inselspital, Bern University Hospital, University of Bern, 3041 Bern, Switzerland; kristina.loessl@insel.ch; 16Department of Radiation Oncology, Cancer Center, University Medical Center Utrecht, 3584 AB Utrecht, The Netherlands; i.m.schulz@umcutrecht.nl; 17Department of Radiation Oncology, Sana Klinikum Offenbach GmbH, 63069 Offenbach, Germany; peter.niehoff@sana.de; 18Department of Oncology, Faculty of Medicine and Health, Örebro University, 70 185 Örebro, Sweden; ruth.sanchez-hermansson@regionorebrolan.se; 19Department of Radiotherapy, Erasmus MC Cancer Institute, 3000 CA Rotterdam, The Netherlands; r.nout@erasmusmc.nl; 20Klinik für Strahlentherapie und Onkologie, Universitätsklinikum Frankfurt, Goethe Universität, 60590 Frankfurt, Germany; nikolaos.tselis@kgu.de

**Keywords:** decision-making, endometrial cancer, brachytherapy, decision tree

## Abstract

**Simple Summary:**

There are various society-specific guidelines addressing adjuvant brachytherapy (BT) after surgery for endometrial cancer (EC). However, these recommendations are not uniform. Against this background, clinicians need to make decisions despite gaps between best scientific evidence and clinical practice. We analysed decision criteria influencing the selection for adjuvant radiotherapy among European radiation oncology experts. For this, GEC-ESTRO provided 19 European radiation oncology experts on gynaecological brachytherapy for decision-making analyses. The manuscript presents patterns in decision-making among these experts and demonstrates areas of consensus/discrepancies. We also analysed dose prescription and techniques of brachytherapy. This analysis is of special value as the objective approach enabled us to obtain an unbiased description of decision-making among the specialists (the study was not aimed to create or enforce a consensus). The manuscript provides valuable insight into clinical decision-making with a high impact on treatment selection, as expected differences between experts were observed. With this manuscript we are able to visualize and quantify these. This information is relevant for interdisciplinary discussions.

**Abstract:**

Background: There are various society-specific guidelines addressing adjuvant brachytherapy (BT) after surgery for endometrial cancer (EC). However, these recommendations are not uniform. Against this background, clinicians need to make decisions despite gaps between best scientific evidence and clinical practice. We explored factors influencing decision-making for adjuvant BT in clinical routine among experienced European radiation oncologists in the field of gynaecological radiotherapy (RT). We also investigated the dose and technique of BT. Methods: Nineteen European experts for gynaecological BT selected by the Groupe Européen de Curiethérapie and the European Society for Radiotherapy & Oncology provided their decision criteria and technique for postoperative RT in EC. The decision criteria were captured and converted into decision trees, and consensus and dissent were evaluated based on the objective consensus methodology. Results: The decision criteria used by the experts were tumour extension, grading, nodal status, lymphovascular invasion, and cervical stroma/vaginal invasion (yes/no). No expert recommended adjuvant BT for pT1a G1-2 EC without substantial LVSI. Eighty-four percent of experts recommended BT for pT1a G3 EC without substantial LVSI. Up to 74% of experts used adjuvant BT for pT1b LVSI-negative and pT2 G1–2 LVSI-negative disease. For 74–84% of experts, EBRT + BT was the treatment of choice for nodal-positive pT2 disease and for pT3 EC with cervical/vaginal invasion. For all other tumour stages, there was no clear consensus for adjuvant treatment. Four experts already used molecular markers for decision-making. Sixty-five percent of experts recommended fractionation regimens of 3 × 7 Gy or 4 × 5 Gy for BT as monotherapy and 2 × 5 Gy for combination with EBRT. The most commonly used applicator for BT was a vaginal cylinder; 82% recommended image-guided BT. Conclusions: There was a clear trend towards adjuvant BT for stage IA G3, stage IB, and stage II G1–2 LVSI-negative EC. Likewise, there was a non-uniform pattern for BT dose prescription but a clear trend towards 3D image-based BT. Finally, molecular characteristics were already used in daily decision-making by some experts under the pretext that upcoming trials will bring more clarity to this topic.

## 1. Introduction

Endometrial cancer (EC) is one of the most common malignancies affecting women in developed countries [1]. Although the majority of cases are diagnosed at an early stage, differences in patient characteristics and histopathological, molecular, and genetic features impact both prognosis and treatment. The primary management of operable EC is total hysterectomy and bilateral salpingo-oopherectomy [2]. The indication for adjuvant radiotherapy (RT) is based on the presence of adverse features for recurrence, such as higher disease stage, increased depth of myometrial invasion, higher histopathological grade, presence of lymphovascular space invasion (LVSI), lymph node involvement, and histologic subtype [2,3,4,5,6,7,8,9]. There is also a development towards decision-making based on molecular features. There are already studies with promising results, e.g., the subgroup analysis of PORTEC III trial [10]. Upcoming trials will hopefully bring more clarity to this topic.

There are two RT modalities implemented in the multidisciplinary treatment of EC: external beam radiotherapy (EBRT) and brachytherapy (BT). Although a meta-analysis showed that postoperative EBRT reduces locoregional recurrence as early as in stage I disease [11], an overall survival benefit could not be demonstrated in randomized trials [12,13,14]. The prospective PORTEC II trial [15] compared vaginal BT and EBRT in patients older than 60 years with stage IB/G1–2 or stage IA/G3 disease. The results showed that both modalities were equieffective in terms of overall survival (OS) and cancer-specific survival with better quality of life in the BT arm. However, the pelvic recurrence rate was lower for EBRT than for BT alone, with 0.5% vs. 3.8% at 5 years, respectively, with EBRT being recommended especially for patients with extensive LVSI, p53 mutation, or L1CAM positivity given the increased risk for pelvic failure [15,16,17]. In this context, the addition of chemotherapy or the replacement of EBRT by chemotherapy has been studied in several randomized trials with benefit in OS and/or disease-free survival (DFS) for some subgroups, e.g., for serous cell histology or for stage III endometrial cancer [18,19,20,21]. Regarding BT, there are various society-specific guidelines addressing its adjuvant implementation for EC [2,3,4,5,6,7,8]. However, these recommendations are not uniform in regard to indication and treatment technique (e.g., dose scheme, prescription depth, target volume). There is also conflict with the clinical practice guidelines advocating for EBRT instead of BT for patients with adverse features for pelvic recurrence. Another limitation of some guidelines is that there are multiple treatment options, allowing for quite a broad range of choices for a given patient. 

Against this backdrop, clinicians often need to make decisions despite gaps between the best scientific evidence and clinical practice. Especially in rapidly evolving fields, the information on what is actually being performed in the radio-oncological community is important; it can facilitate the generation and distribution of knowledge by adding information to the classical approach of evidence-based medicine, which is the most important in clinical settings where multiple treatment options can be offered [22]. EC is a characteristic example where multiple factors affect the adjuvant strategy, and further in-depth information on daily clinical decision-making—often complex—seems prudent. This information can be optimally obtained in the form of decision trees, which allow the analysis of multiple clinically implemented strategies [23].

Hence, the aim of this work was to identify which disease characteristics are essential for the decision-making process in the adjuvant application of RT for EC. We wanted to provide insight into real-life decision-making among European experts and capture the various decision criteria and treatment strategies while evaluating consensus and dissent. This may help in identifying problems in the implementation of guidelines and to be also considered in future recommendations.

## 2. Methods

We asked radiation oncologists across Europe who were selected as experts in the field of gynaecological cancers by the Groupe Européen de Curiethérapie and the European Society for Radiotherapy & Oncology (GEC-ESTRO gynaecological network group) to answer the following questions: “Please describe for which patients with R0 resected endometrial cancer you would recommend brachytherapy? If you use brachytherapy, please describe your dose and technique (e.g., prescription depth, image-based or not, dose restrictions for OARs). Please describe any criteria used in your decision-making”. Answers were allowed in any format (e.g., free text, tables, diagrams, or figures). No specific clinical scenarios, examples, or decision criteria were proposed to avoid influencing responses. Decision parameters were identified, and a list of unified decision criteria was determined. The final decision trees were analysed with a methodology based on diagnostic nodes, which allows for an automated cross-comparison of decision trees [24]. Consensus and discrepancies were evaluated with the objective consensus methodology as previously described [23,25]. This approach was developed by Putora et al. and has already been implemented in several similar projects [26,27,28,29,30]. Data collection of the different centers was carried out by bilateral contact of the coordinator (MG) and the participating experts. The resulting decision trees were presented to the participants for verification and modifications, if considered necessary. The most common parameters for decision-making were determined and converted into decision trees, and areas of discordance were identified.

To enable cross-comparison of algorithms, compatible criteria are a prerequisite. Similar decision criteria were fused into uniform categories. For example, some of our experts used the FIGO stage, others used the TNM classification and still others used prognostic risk groups for decision-making. For better comparison, we decided to use the TNM stage. For this, we broke down FIGO stages and risk groups into individual factors. When more than 60% of the experts recommended the same treatment, we defined it as a consensus (representing a majority).

For this project, no patient or preclinical data were collected.

## 3. Results

Twenty-two radiation oncologists from different institutions with expertise in the field of gynaecological cancers and recommended by the GEC-ESTRO gynaecological network were invited for this analysis. Nineteen experts took part in the analysis, while three experts did not respond to repeated requests and thus were excluded from data collection.

Figure 1 shows an example of an individual decision tree. The main decision criteria used by our experts were tumour extension (pT), grading (G1/2, G3), nodal status (pN0, pN+, pNx/cN0), lymphovascular invasion (LVSI–extensive: yes/no), and cervical stroma/vaginal invasion (yes/no). For a better overview of the decision trees, we excluded decision criteria only mentioned by a minority of experts (i.e., insular criteria) [28]. Only four of the experts (already) used molecular tumour features for decision-making in routine clinical practice. Age, performance status, and/or comorbidities were mentioned by four experts as relevant decision criteria but were excluded from the final decision trees for better comparability. The ability to attend follow-up visits (psychosocial factors, residency, mobility) was also mentioned as a decision criterion by one expert, and this factor was also excluded from the analysis.

For simplification, non-endometrioid histologies were excluded from our analysis, and the role of chemotherapy was not addressed.

For two experts, BT and/or EBRT played no role in the adjuvant setting of resected endometrial cancer with clear resection margins. For all other experts, RT (BT alone, EBRT + BT, EBRT alone) was used in the adjuvant setting, except for pT1a G1–2 N0 without extensive LVSI, where there was 100% agreement among all experts not to perform adjuvant RT. The consensus for adjuvant BT or EBRT + BT is shown in Figure 2. For pT1a G3 LVSI-negative EC, the majority of experts (84%) recommended BT. In the case of extensive LVSI, the recommendation rate decreased to 58% as more experts then recommended EBRT. The same trend was demonstrated for pT1b N0 disease. For 74% of the experts, adjuvant BT was the treatment of choice for pT1b G1-2 LVSI-negative, 63% decided for adjuvant BT for pT1b G3 LVSI-negative, 74% of experts used BT for pT2 G1, and 58% for pT2 G2 LVSI-negative disease. For pT2 N0 G3, up to 58% of experts recommended EBRT + BT, and seven (37%) experts used only BT, especially for patients without extensive LVSI. One expert explicitly mentioned that for pT2 G1-2 nodal negative disease with only microscopic stromal invasion adjuvant BT is the treatment of choice. Up to 84% of experts used EBRT + BT for nodal positive pT2 disease and 79–84% for pT3 endometrial cancer with cervical/vaginal invasion. For pT4 there was no consensus. An overview of the recommendations for each stage is shown in Figure 3, Figure 4 and Figure 5.

Fractionation schemes, dose prescription, and planning specifications are shown in Table 1. In summary, for adjuvant BT as monotherapy, 3 × 7 Gy or 4 × 5 Gy were the most commonly prescribed regimens. For the combination with EBRT, 65% of experts who saw an indication for BT recommended 2 × 5 Gy. The most commonly used applicator for BT was a vaginal cylinder. The dose was prescribed by 76% of experts to the upper third or upper 3 cm of the vagina. Ninety-five percent of the experts prescribed the dose to a depth of 5 mm from the applicator surface. Eighty-two percent recommended image-guided BT. The constraints for organs at risk (OAR) varied among the experts. Ten experts did not mention specific restrictions for OARs for BT without EBRT. For all other experts, the constraints for the bladder, rectum and sigmoid are listed in Table 1.

## 4. Discussion

There are different society-specific guidelines addressing the role of adjuvant radiotherapy for endometrial cancer. However, these recommendations are not uniform in regard to indication and treatment technique. This is the first report comparing treatment algorithms in the adjuvant setting of EC using objective consensus methodology. We performed this analysis among very experienced radiation oncologists in the field of gynaecological cancers across Europe. The experts were appointed by the chairs of the GEC-ESTRO gynaecological network group based on their expertise in the field of gynaecological BT. All selected experts are working in a leading position in high-volume gynecological cancer centres for brachytherapy for several years. 

The analysis consisted of an open question, which is very relevant to the methodology as we wanted to avoid pre-defining clinical scenarios or specific criteria. We only analyzed decision criteria mentioned by the experts. Among the participating peers, we found similar decision criteria used in the adjuvant management of EC, which were in accordance with the most recent guidelines [2,3,4,5,6,7,8,9]. However, the criteria were implemented differently. For many scenarios, there was no clear consensus among the experts. Adjuvant treatment was not recommended for pT1a G1–2 disease, defined as low-risk EC, which is in accordance with the literature [11,31]. We observed a consensus for BT for EC in stage IA G3 and IB G1–3 and for stage II G1 without LVSI. External beam radiotherapy +/− BT was recommended for more advanced stages or in patients with extensive LVSI. This heterogeneity of treatment recommendations is also reflected in international guidelines [2,3,4,5,6,7,8]. This is particularly striking for nodal negative but LVSI positive endometrial cancer. For this situation there was no consensus among the experts, especially for pT1 disease. While some recommended BT alone, some preferred EBRT, and others a combination of EBRT + BT. There were also some experts who did not recommend adjuvant treatment. However, the higher the grading, the more experts recommended adjuvant treatment. At this, substantial LVSI and poorly differentiated histology are two of the most relevant risk factors for locoregional recurrence [32]. The most recent ESGO/ESTRO/ESP guideline suggested adjuvant EBRT to be considered for stage I EC with substantial LVSI (especially those patients with G3 grading or omitted surgical lymph node staging) [6]. PORTEC I and II showed that patients with substantial LVSI who received no adjuvant treatment or vaginal BT had a 5-year risk for pelvic regional recurrence of 25–30%, which was reduced to 5% by the addition of EBRT. It must be noted that only 5% of the patient cohorts in those two studies had substantial LVSI. As we have no significant survival difference in this specific cohort between BT or EBRT, and studies of accordingly selected patients for this entity are missing, the choice of treatment is often based on expert opinions as illustrated in this analysis. One may choose BT to avoid radiation toxicity, others may decide for EBRT to improve pelvic control but with non-significant survival benefit [33].

For most experts who recommended EBRT for more advanced disease stages, there was also an indication for additional BT. However, the role of vaginal BT in combination with EBRT is not clear except for R1 resected tumours. Bingham et al. evaluated 12,988 patients with stage III EC and demonstrated that the addition of BT to EBRT significantly affected survival among women with endocervical or cervical stromal invasion [34]. An older SEER analysis also indicated that the addition of a BT boost to EBRT is of benefit in patients with stage IIIC disease with extension of the tumour beyond the uterus [35]. In contrast, there are several studies showing no difference in local recurrence or OS rates by adding BT in patients experiencing cervical invasion while pointing out the increased risk of toxicity [36,37,38,39]. The lack of agreement between the experts in these aspects is related to the lack of strong evidence in recent prospective studies. Therefore, it seems there is the need for more multicentre-specific analysis to unify the criteria.

Another treatment strategy for resected endometrial cancer with clear resection margins recommended by two of our experts was close surveillance and salvage RT for local recurrence. They argued that RT was found to be a very effective salvage treatment for local recurrences in not previously irradiated patients [40,41]. Many of the experts participating in this analysis made their decisions based on the risk group classification described in the PORTEC I [12] and GOG 99 trials [13] or based on the consensus published by ESMO/ESGO/ESTRO [2]. The latter was used by the majority of the experts. The most recent ESGO/ESTRO/ESP guidelines [6] for the management of patients with EC recommend the additional use of molecular features for classification, if known. Only a few of the experts in our analysis routinely used molecular classification for decision-making. However, in the discussion with the experts, there was a clear trend towards implementation of molecular decision criteria for daily routine in the near future. Nevertheless, most of the experts used molecular markers only within clinical trials. One of these trials is the ongoing PORTEC 4a study, a multicentre international phase III randomized trial of molecular-integrated risk profile-based adjuvant treatment (experimental arm) or adjuvant vaginal BT (standard arm) in women with high-intermediate risk EC [42]. From the 10-year results of the PORTEC II trial, L1CAM and p53-mutant expression and substantial LVSI were risk factors for pelvic recurrence and distant metastasis [17]. This was also confirmed by a systematic review [43] and other retrospective studies [44,45]. Interestingly, none of the experts mentioned L1CAM for decision-making in routine clinical practice. The most relevant factors for the majority of the experts were tumour extension, lymph node status, and the extent of LVSI. In the literature, there are also other factors used for decision-making, e.g., age. In the PORTEC I [12] and GOG 99 trials [13], patient age was included for risk classification. The NCCN Guidelines (Version 3.2021) recommend using age for decision-making, while in other guidelines, patient age no longer directly impacts decision-making [4,6]. The most recently published ESGO/ESTRO/ESP guideline mentioned age for decision-making only for intermediate risk groups: “The omission of adjuvant brachytherapy can be considered, especially for patients aged < 60 years”. However, only a minority of the experts participated in our analysis used age as a decision-making factor. Two of the experts explicitly mentioned age as an important factor for decision-making, while other experts mentioned age but did not define it as a strict criterion. It was listed as a factor by four experts along with patient preference to choose between follow-up vs. BT for stage IA G3 or IB G1–2 disease.

A commonly used fractionation regimen for adjuvant BT as monotherapy is 7 Gy for three fractions (total physical dose of 21 Gy) prescribed at 5 mm depth from the applicator surface (NCCN Version 3.2021). This regimen was also used in the PORTEC II trial. In combination with EBRT, the most common scheme described in the literature is 2 × 5 Gy prescribed at 5mm depth from the applicator surface, which was also the recommended regimen in the PORTEC III trial (11). As comparative studies of dose-fractionation schemes are scarce, the clinically implemented regimes often depend on local standards and the experience of each centre [46]. This was also demonstrated in our analysis where nine different regimens were reported for BT as monotherapy, and six in the combination with EBRT. The EQD2 (α/β = 10) ranged from 19.5 Gy–37.5 Gy for BT as monotherapy and from 9.9 Gy–18.8 Gy for BT in combination with EBRT. Furthermore, the majority of experts (71%) administered the treatment to the upper third of the vagina using a commercially available afterloading vaginal cylinder. There was also consensus prescribing the dose to 5 mm depth from the applicator surface. While the use of three-dimensional image-guided BT (3D-IGBT) is highly recommended in the treatment of cervical cancer [47,48,49], in EC, many centres perform two-dimensional BT using standard plans. This technique follows a “one-size-fits-all” approach and does not allow for individualized dose optimization based on patient-specific parameters. There are some reports indicating that 3D treatment planning for vaginal BT improves target coverage by identifying air gaps between the vaginal mucosa and the applicator [50,51,52], with other authors suggesting that smaller air gaps (<0.7 cm [3]) do not increase the risk for vaginal cuff failure [53]. Overall, only limited data exist for 3D-IGBT in vaginal cuff treatment, and further studies are needed to elucidate whether this evolution of the BT technique can improve outcomes and reduce late toxicity in the adjuvant setting. However, in our analysis, we showed that two-thirds of the experts had already implemented 3D-IGBT using either CT-, US- or MRI-guided techniques although most of them are prescribing the dose to 5 mm from the vaginal cylinder. The main arguments next to optimization of dose distribution were the possibility to identify a cuff perforation by the applicator, recognize a correct contact of the applicator to the mucosa, or to adapt the treatment scheme, if necessary, considering OAR dose constraints and the dose administered to the vaginal wall [54]. Currently the dose is prescribed to the applicator surface or at 5mm, and a more modern approach could be the prescription to a vaginal CTV establishing the best dose and fractionation to be administered in future studies.

The limitations of our study include patient heterogeneity and case complexity, as encountered in the clinical routine of every individual expert, which cannot be addressed for general consideration among all experts. At the same time, since many experts in this study are involved in ongoing (and potentially the same) clinical trials, there is a possibility of bias in their recommendations. In addition, as the role of chemotherapy in the adjuvant setting remains controversial with no uniform approach concerning its implementation, we plan to focus on this topic through a separate decision-making project which will not cover the role of specific radiotherapy modalities. Adding this question to the recent project would have made the analysis too complex given the intricacy of the underlying question. Notwithstanding, the aspect of adjuvant chemotherapy represents an ongoing topic and its indication should be evaluated within the framework of interdisciplinary tumour conferences which take full account of input from the gynecological as well as medical oncology point of view. The indication of adjuvant RT must likewise be evaluated accordingly. We also did not address treatment strategies for non-endometrioid histologies. Finally, we used a reduced number of simplified decision criteria for better visualization of the decision trees, taking into account the risk of not depicting every individual treatment decision.

## 5. Conclusions

The objective consensus approach, used in this study, enabled us to obtain an unbiased description of decision-making among experienced radiation-oncologists in the field of gynaecological cancer. In summary, RT is routinely performed in the adjuvant treatment of EC except for pT1a G1–2 LVSI-negative disease. There was a clear trend towards adjuvant BT for node-negative stage pT1a G3, pT1b G1–3, or pT2 G1–2 disease without LVSI. However, there was also a variation in the interpretation of risk factors by the individual experts, leading to a variation in adjuvant treatment recommendations (BT vs. EBRT+/−BT). Moreover, the reported dose-fractionation schemes were not uniform. Despite the good results of 2D planning, there was a clear trend towards image-guided BT that will likely be developed further in the future. We also showed that molecular characteristics are already used in decision-making by few experts, with upcoming trials hopefully bringing clarity to this topic. The generated information of this study may be of relevance for further interdisciplinary discussions on good clinical practice as well as potential adaptions at the guideline level.

## Figures and Tables

**Figure 1 cancers-14-00906-f001:**
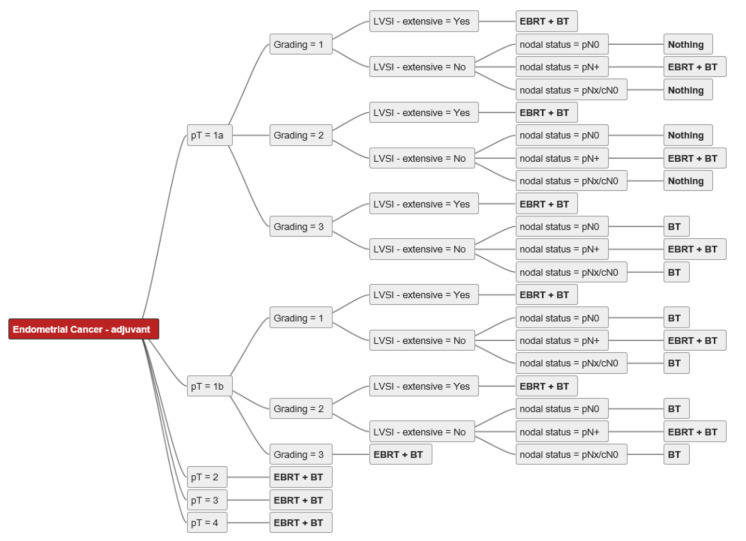
Individual decision tree from expert A. Abbreviations: pT–pathological tumour stage; pN0-pathological nodal negative disease; pN+-pathological nodal positive disease; pNx/cN0–no pathological nodal staging, but clinical negative nodal staging; LVSI–lymphovascular space invasion; EBRT–external beam radiotherapy; BT-brachytherapy; bold-treatment strategies.

**Figure 2 cancers-14-00906-f002:**
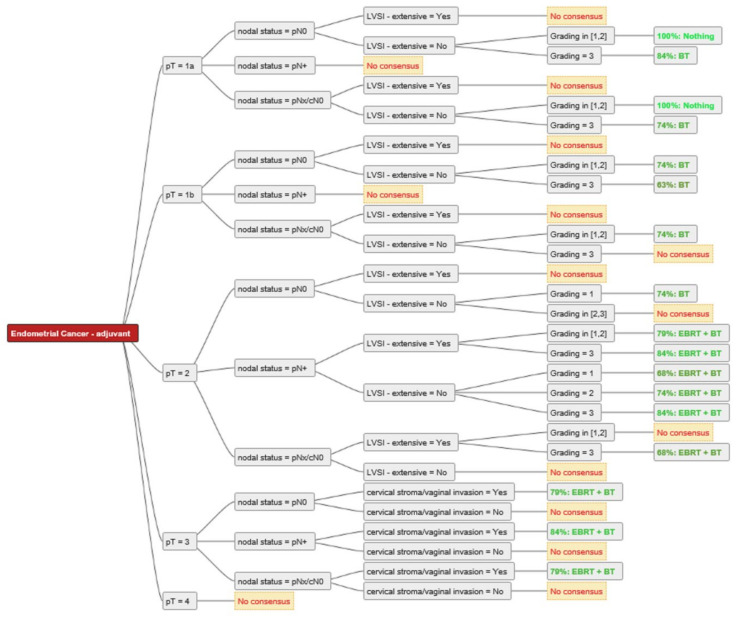
Consensus for adjuvant treatment strategies among the experts. Consensus was defined as ≥60% rate of recommendation. Abbreviations: pT–pathological tumour stage; pN0-pathological nodal negative disease; pN+-pathological nodal positive disease; pNx/cN0–no pathological nodal staging, but clinical negative nodal staging; LVSI–lymphovascular space invasion; EBRT–external beam radiotherapy; BT–brachytherapy; Nothing–no adjuvant external beam radiotherapy and/or brachytherapy.

**Figure 3 cancers-14-00906-f003:**
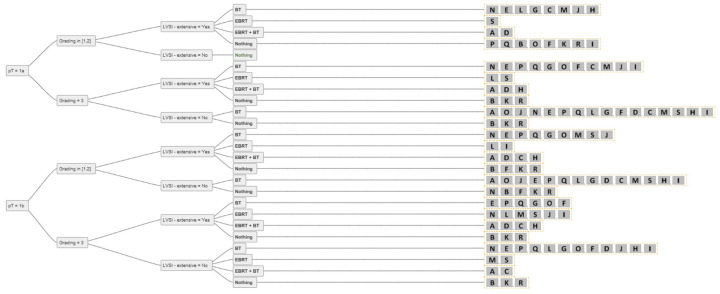
The answers from all experts (A-S) for all combinations of simplified decision criteria for nodal-negative pT1 endometrial cancer. Abbreviations: pT–pathological tumour stage; LVSI–lymphovascular space invasion; EBRT–external beam radiotherapy; BT–brachytherapy; Nothing–no adjuvant external beam radiotherapy and/or brachytherapy.

**Figure 4 cancers-14-00906-f004:**
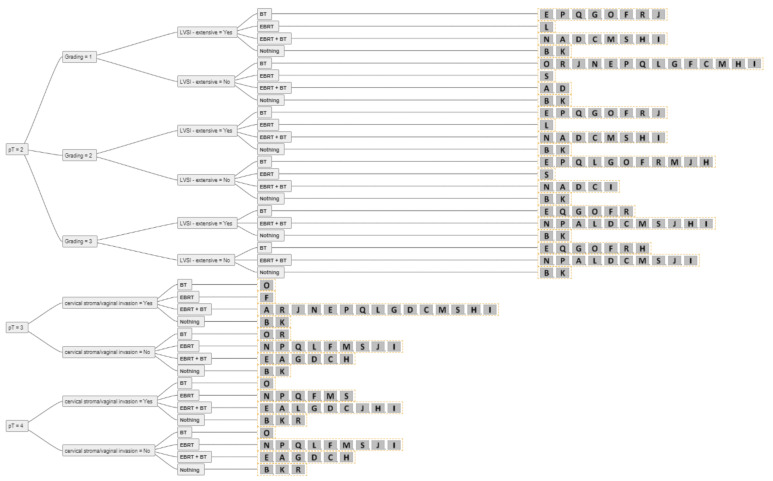
The answers from all experts (A-S) for all combinations of simplified decision criteria for nodal-negative pT2-4 endometrial cancer. Abbreviations: pT–pathological tumour stage; LVSI–lymphovascular space invasion; EBRT–external beam radiotherapy; BT–brachytherapy; Nothing–no adjuvant external beam radiotherapy and/or brachytherapy.

**Figure 5 cancers-14-00906-f005:**
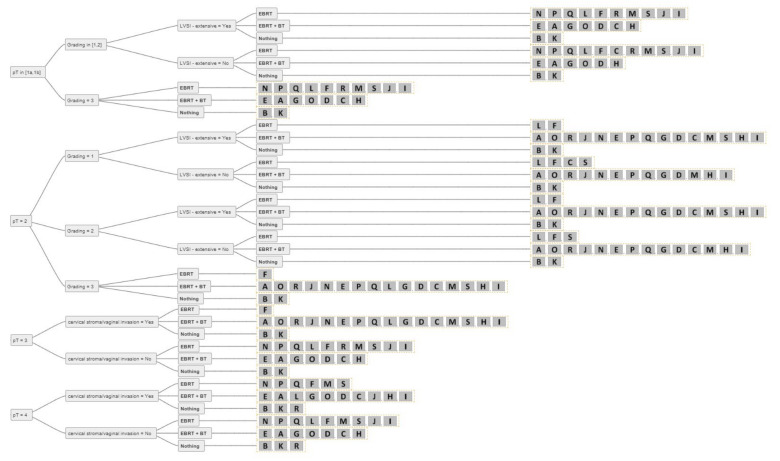
Answers from all experts (A-S) for all combinations of simplified decision criteria for all stages of nodal-positive endometrial cancer. Abbreviations: pT–pathological tumour stage; LVSI–lymphovascular space invasion; EBRT–external beam radiotherapy; BT–brachytherapy; Nothing–no adjuvant external beam radiotherapy and/or brachytherapy.

**Table 1 cancers-14-00906-t001:** Brachytherapy–dose prescriptions and planning details, recommended by our experts (A-R).

Expert	BT-Dose (without EBRT)	Fractions per Week	BT-Dose (Combined with EBRT)	Fractions per Week	Prescription	Applicator	Image-Based (CT, MRI, US)	OAR Constraints
A	2 × 7.5 Gy EQD2(10 Gy) = 21.9 Gy EQD2(3 Gy) = 31.5 Gy	2	1 × 7 Gy EQD2(10 Gy) = 9.9 Gy EQD2 (3 Gy) = 14.0 Gy	1	5 mm from applicator surface, upper 3 cm of vagina	vaginal cylinder	yes	doses to D2cc OAR never are superior than the inferior limits of dose of Embrace; 68 Gy EQD2 (α/β = 3) dose constraint at 2cm3 of the most exposed area to the dose of the vaginal CTV
B	Do not provide adjuvant brachytherapy for R0 resected endometrial cancer							
C	3 × 7 Gy EQD2(10 Gy) = 29.8 Gy EQD2(3 Gy) = 42.0 Gy	1	2 × 5 Gy EQD2(10 Gy) = 12.5 Gy EQD2(3 Gy) = 16.0 Gy	1–2	5 mm depth, upper third of vagina	vaginal cylinder	yes	only for EBRT + BT: D2cc Bladder < 80 Gy, D2cc Rectum < 65 Gy, D2cc Sigmoid/Small Bowel < 70 Gy (α/β = 3)
D	4 × 6Gy EQD2(10 Gy) = 32.0 Gy EQD2(3 Gy) = 43.2 Gy	1	2–3 × 5 Gy EQD2(10 Gy) = 12.5–18.8 Gy EQD2(3 Gy) = 16.0–24.0 Gy	1	5 mm from applicator surface; upper 3 cm of vagina	vaginal cylinder	yes	only for EBRT + BT: Small bowell < 70 Gy (sum of Dmax EBRT and D2cc BT); Rectum < 70 Gy (sum of Dmax EBRT and D2cc BT); Bladder < 90 Gy (sum of Dmax EBRT and D2cc IRT)
E	4 × 5 Gy EQD2(10 Gy) = 25.0 Gy EQD2(3 Gy) = 32.0 Gy	1–2	2 × 5 Gy EQD2(10 Gy) = 12.5 Gy EQD2(3 Gy) = 16.0 Gy	1–2	5 mm depth, upper third of vagina	vaginal mold applicator, If parametrial involvement or margin or vaginal involvement: 15 Gy PDR, one fraction, 0.5 Gy per pulse	yes	only for EBRT + BT: bladder, rectum, and sigmoid colon D2cc < 70, 65, and 65 Gy (α/β = 3)
F	6 × 5 Gy EQD2(10 Gy) = 37.5 Gy EQD2(3 Gy) = 48 Gy	2	2 × 5 Gy EQD2(10 Gy) = 12.5 Gy EQD2(3 Gy) = 16.0 Gy	2	2–3 mm depth, upper third of vagina	intracavitary technique, individual diameter of applicator	yes	OAR- rectum and bladder–calculation of max. dose for rectum and bladder based on individual measurement of closest distance between vaginal surface and rectum ant. wall and bladder post. wall–using high resolution transvaginal ultrasound (if necessary also TRUS).
G	4 × 5.5 Gy EQD2(10 Gy) = 28.4 Gy EQD2(3 Gy) = 37.4 Gy	1	2 × 5 Gy EQD2(10 Gy) = 12.5 Gy EQD2(3 Gy) = 16.0 Gy	1	5 mm depth from applicator surface, upper third of vagina	vaginal cylinder; if parametrial/paravainal involvement: interstitial needles	yes	rectum/bladder point dose <110% of prescription dose. If point dose >110% of dose we decrease the prescription dose from 5.5 Gy to 5.0 Gy.
H	4 × 5 Gy EQD2(10 Gy) = 25.0 Gy EQD2(3 Gy) = 32.0 Gy Or 3 × 7 Gy EQD2(10 Gy) = 29.8 Gy EQD2(3 Gy) = 42.0 Gy	2–3	2 × 6 Gy EQD2(10 Gy) = 16.0 Gy EQD2(3 Gy) = 21.6 Gy	1–2	5 mm depth, upper third of vagina	vaginal cylinder	yes	no specific restrictions
I	4 × 5 Gy EQD2(10 Gy) = 25.0 Gy EQD2(3 Gy) = 32.0 Gy	2	2 × 5 Gy EQD2(10 Gy) = 12.5 Gy EQD2(3 Gy) = 16.0 Gy	2	5 mm from applicator surface, upper third of vagina	vaginal cylinder	no	no specific restrictions
J	6 × 4.5 Gy EQD2(10 Gy) = 32.6 Gy EQD2(3 Gy) = 40.5 Gy	3	3 × 4 Gy EQD2(10 Gy) = 14.0 Gy EQD2(3 Gy) = 16.8 Gy	3	5 mm from applicator surface, upper third of vagina (3 cm)	vaginal cylinder	no	no specific restrictions
K	Do not provide adjuvant brachytherapy for R0 resected endometrial cancer							
L	5 × 5 Gy EQD2(10 Gy) = 31.3 Gy EQD2(3 Gy) = 40.0 Gy	2	2 × 5 Gy EQD2(10 Gy) = 12.5 Gy EQD2(3 Gy) = 16.0 Gy	2	5 mm depth, target volume: half of vaginal lenght minus 1cm	vaginal cylinder	no	rectal dose <5 Gy per fraction (in vivo dosimetry with rectal probe: if dosis exceeds 5 Gy, CT-based BT is used for the remaining fractions)
M	4 × 5 Gy EQD2(10 Gy) = 25.0 Gy EQD2(3 Gy) = 32.0 Gy	2	2 × 5 Gy EQD2(10 Gy) = 12.5 Gy EQD2(3 Gy) = 16.0 Gy	2	5 mm depth from vaginal mucosa, upper third of vagina	vaginal cylinder	yes	no specific restrictions
N	3 × 7 Gy EQD2(10 Gy) = 29.8 Gy EQD2(3 Gy) = 42.0 Gy	2	na	na	5 mm from applicator surface; proximal 3.5–4 cm of the vagina	generally vaginal cylinder, seldom ovoids	yes	OAR delineated up to at least 2 cm cranially from the applicator and constrains are for the bladder D2cc < 7,5 Gy, for the rectum and the bowel D2cc < 7 Gy per fraction.
O	3 × 7 Gy EQD2(10 Gy) = 29.8 Gy EQD2(3 Gy) = 42.0 Gy	1	2 × 5 Gy EQD2(10 Gy) = 12.5 Gy EQD2(3 Gy) = 16.0 Gy	2	5 mm depth, upper third of vagina	vaginal cylinder	yes	no specific restrictions
P	3 × 7 Gy EQD2(10 Gy) = 29.8 Gy EQD2(3 Gy) = 42.0 Gy	1	2 × 6 Gy EQD2(10 Gy) = 16.0 Gy EQD2(3 Gy) = 21.6 Gy	1–2	5 mm depth, 2–4 cm of proximal vagina	vaginal cylinder	yes	no specific restrictions
Q	4 × 5 Gy EQD2(10 Gy) = 25.0 Gy EQD2(3 Gy) = 32.0 Gy	2	2 × 5 Gy EQD2(10 Gy) = 12.5 Gy EQD2(3 Gy) = 16.0 Gy	2	5 mm depth, upper third of vagina (3 cm)	vaginal cyinder	yes	rectal dose <5 Gy per fraction
R	6 × 3 Gy EQD2(10 Gy) = 19.5 Gy EQD2(3 Gy) = 21.6 Gy	5	4 × 2.5 Gy EQD2(10 Gy) = 10.4 Gy EQD2(3 Gy) = 11.0 Gy	5	upper 2/3 of the vagina with a 5 mm dose specification depth from the vaginal surface.	vaginal multichanel cilinder (VMC).	yes	no specific restrictions
S	3 × 7 Gy EQD2(10 Gy) = 29.8 Gy EQD2(3 Gy) = 42.0 Gy	1	2 × 5 Gy EQD2(10 Gy) = 12.5 Gy EQD2(3 Gy) = 16.0 Gy	2	5 mm depth, upper third of vagina; in combination with EBRT: 5 mm depth, 2 cm of proximal vagina	vaginal cylinder	yes	PORTEC-4a

Abbreviations: BT–Brachytherapy, CT-Computed tomography, D2cc–dose in 2 cubic centimeter volume, EQD2-equivalent dose in 2 Gy fraction, Gy–Gray, MRI-magnetic resonance imaging, n.a–not applicable, OAR–organ at risk, US–ultrasound, PDR–Pulsed Dose Rate.

## Data Availability

The data presented in this study are available on request from the corresponding author.

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
