# Peer review of "Role of Brachytherapy in the Postoperative Management of Endometrial Cancer: Decision-Making Analysis among Experienced European Radiation Oncologists"

_cancers, 2022, doi:10.3390/cancers14040906_

Round 1
Reviewer 1 Report
The authors investigate about the role of brachytherapy in the postoperative management of endometrial cancer and the aim of this paper is to identify which disease characteristics are essential for the decision-making process in the adjuvant application of radiation therapy for endometrial cancer. Although there are various society-specific guidelines addressing adjuvant implementation for endometrial cancer with brachytherapy, these recommendations are not uniform in regard to indication and treatment technique and there are also multiple treatment options, allowing for quite a broad range of choices for a given patient.
The authors explored factors influencing decision-making for adjuvant brachytherapy in clinical routine among experienced European radiation oncologists in the field of gynaecological radiotherapy and their results show a variation in the interpretation of risk factors by the individual experts, leading to a variation in adjuvant treatment recommendations (BT vs. EBRT+/-BT). Also, the reported dose-fractionation schemes were not uniform.
In this report the final decision trees were analysed with a methodology based on diagnostic nodes, which allows for an automated cross-comparison of decision trees. Consensus and discrepancies were evaluated with the objective consensus methodology.
The manuscript is well written, the findings of this study could help improve the new guidelines and could be incorporated into future recommendations.
The reader is given an adequate background about the topic through a careful literature review.
The English language is good, the figures and table are clear, the reference list covers the relevant literature adequately.
Author Response
Thank you for your feedback.
Reviewer 2 Report
This manuscript described the Decision-making analysis for Role of Brachytherapy in the postoperative management of endometrial cancer. This is a very interesting report in this field and an important study for the treatment of endometrial cancer. There are some useful consensus findings in this study. However, there are several problems in this manuscript and I do not think this manuscript is worth publishing in Cancers.
- There is no mention of chemotherapy in this paper. There is a great potential for differences in thinking depending on whether chemotherapy is indicated or not.
- The strategy of postoperative adjuvant therapy may vary depending on the gynecological surgical technique of gynecologist in each fasilities. This point has not been taken into account at all.
- It is necessary to indicate how the experts who participated in this analysis were selected, what their years of experience and number of experienced cases are, and whether there is any bias that might affect this analysis. Since many experts in this study are involved in ongoing clinical trials, there is a possibility of bias from participation in clinical trials.
Author Response
There is no mention of chemotherapy in this paper. There is a great potential for differences in thinking depending on whether chemotherapy is indicated or not.
Answer: As the role of chemotherapy in the adjuvant setting is still controversial with no uniform approach concerning its implementation, we decided, in accordance with the understanding of all participating experts, to focus on this topic through a separate decision-making project which will not focus on specific radiotherapy modalities. Adding this question to the recent project would make the analysis too complex given the intricacy of the underlying radiotherapy question. Therefore, we decided after careful consideration to focus only on radiotherapy approaches.
The strategy of postoperative adjuvant therapy may vary depending on the gynecological surgical technique of gynecologist in each fasilities. This point has not been taken into account at all.
Answer: The analysis consisted of an open question: “Please describe for which patients with R0 resected endometrial cancer you would recommend brachytherapy?”. This is very relevant to the methodology as we wanted to avoid pre-defining and limiting clinical scenarios or specific criteria. As a result, we received various decision-criteria. As the surgical technique was not mentioned by the experts, this factor was not included in the manuscript. However, we added this information to the Discussion section in order to address this aspect: “The analysis consisted of an open question. This is very relevant to the methodology as we wanted to avoid pre-defining clinical scenarios or specific criteria. We only analyzed decision criteria mentioned by the experts.”
It is necessary to indicate how the experts who participated in this analysis were selected, what their years of experience and number of experienced cases are, and whether there is any bias that might affect this analysis. Since many experts in this study are involved in ongoing clinical trials, there is a possibility of bias from participation in clinical trials.
Answer: The participating experts were suggested by the GEC-ESTRO gynaecological network group. In fact, all selected radiation oncologists have substantial experience in the field of gynaecological brachytherapy both within the framework of their respective national radiotherapy societies as well as active members of GEC-ESTRO working groups. All participating peers are working in widely recognized high-volume centres for brachytherapy in leading positions. This ancillary information was added to the manuscript. We also added the following sentence to the limitations of this study: “Since many experts in this study are involved in ongoing (and potentially the same) clinical trials, there is a possibility of bias in their recommendations.”
Reviewer 3 Report
The article investigated different practices and factors on decision making about the use of brachytherapy in endometrial cancer (EC).
The results from this work reflected the real-world situation. Despite the ESMO-ESTRO-ESGO guideline, the use of radiotherapy (RT) varied across different centers. There were multiple parameters like age, extent of lymphadenectomy, histology including grading, stage, molecular markers that might alter the decision on the choice of RT.
The abstract summarized those important points. However, it would be clearer if it could specify whether the BT was used alone, or with external RT under different scenarios.
Line 82: Most hysterectomy is now done by laparoscopic or robot-assisted route, not abdominal route. Suggest to delete the word ‘abdominal’.
Methodology: The use of molecular reporting system by pathologists, the extent of operation, e.g., circumstance where lymphadenectomy (or sentinel lymph node biopsy) and omentectomy would be performed, were not discussed. As stated in Lines 177 and 333, the use of chemotherapy was not included, which was often used together with RT in advanced EC. Another drawback was there was no correlation with data on recurrence.
The decision-making trees were somehow difficult to read.
Besides, the rationale of the administration and protocol of BT among these radiation oncologists was only partially discussed. For example, whether the extent of operation or the use of chemotherapy would alter their choice of RT, except very briefly in Line 289. The complication rates of the different BT protocols were not elaborated.
And as the implementation of molecular classification is getting more popular, more physicians would base on it to triage the use of adjuvant therapy. For example, early-stage EC patients with POLE mutation may not need adjuvant therapy. Hence, the implication of the results from this work may not be significant in near future.
Author Response
The article investigated different practices and factors on decision making about the use of brachytherapy in endometrial cancer (EC).
The results from this work reflected the real-world situation. Despite the ESMO-ESTRO-ESGO guideline, the use of radiotherapy (RT) varied across different centers. There were multiple parameters like age, extent of lymphadenectomy, histology including grading, stage, molecular markers that might alter the decision on the choice of RT.
The abstract summarized those important points. However, it would be clearer if it could specify whether the BT was used alone, or with external RT under different scenarios.
Answer: Whether BT was used alone or in combination with EBRT has already been mentioned in the Abstract. Hereunto, EBRT + BT was the treatment of choice for nodal-positive pT2 disease and for pT3 endometrial cancer with cervical/vaginal invasion, recommended by 74-84% of experts. For other stages BT alone was the preferred treatment.
Line 82: Most hysterectomy is now done by laparoscopic or robot-assisted route, not abdominal route. Suggest to delete the word ‘abdominal’.
Answer: Thank you for this input. We agree and deleted “abdominal”.
Methodology: The use of molecular reporting system by pathologists, the extent of operation, e.g., circumstance where lymphadenectomy (or sentinel lymph node biopsy) and omentectomy would be performed, were not discussed. As stated in Lines 177 and 333, the use of chemotherapy was not included, which was often used together with RT in advanced EC. Another drawback was there was no correlation with data on recurrence.
Answer: Our analysis consisted of an open question: “Please describe for which patients with R0 resected endometrial cancer you would recommend brachytherapy?”. This is very relevant to the methodology as we wanted to avoid pre-defining clinical scenarios or specific criteria. As the molecular reporting system by pathologists or the extent of operation/surgical technique was not named by the experts, these specific aspects could not be analyzed and reflected in the manuscript. As the role of chemotherapy in the adjuvant setting is still controversial with no uniform approach concerning its implementation, we decided, in accordance with the understanding of all participating experts, to focus on this topic through a separate decision-making project which will not focus on specific radiotherapy modalities. Adding this question to the recent project, along with the topic of recurrence management, would make the current analysis too complex and impair significantly the clear interpretation of the trees. Therefore, we decided after careful consideration to focus only on radiotherapy approaches.
The decision-making trees were somehow difficult to read.
Besides, the rationale of the administration and protocol of BT among these radiation oncologists was only partially discussed. For example, whether the extent of operation or the use of chemotherapy would alter their choice of RT, except very briefly in Line 289. The complication rates of the different BT protocols were not elaborated.
And as the implementation of molecular classification is getting more popular, more physicians would base on it to triage the use of adjuvant therapy. For example, early-stage EC patients with POLE mutation may not need adjuvant therapy. Hence, the implication of the results from this work may not be significant in near future.
Answer: As already mentioned in our manuscript we likewise think that in the near future decision-making will be more and more based on molecular features. However, in many centres not all (relevant) molecular analyses are available/offered in full extent yet. Indubitably, in the near future this instrument will be implemented comprehensively but is currently not of sustained relevance in daily routine. From that point of view, our manuscript gives a genuine insight into daily decision-making while representing a solid basis for additional analyses. In fact, we are planning a follow-up project evaluating the trend towards decision-making based on molecular features. This project is intended for next year in consideration of a transition period which seems further necessary for comprehensive adaptions in clinical practice on the European level.
Reviewer 4 Report
This was an excellent analysis of the current practices of adjuvant radiotherapy for endometrial cancer.
Looking at figure 2 (which was most interesting), one does not know how to deal with LVSI positive and node negative patients. The reason for this indecision or lack of consensus should be discussed. My understanding of this/these group of patients is that if we lump stage 1 patients with endometrioid histology, any grade, and any myometrial invasion in one group, we will find any failure rates of 19% and 14% if treated by vault brachytherapy (VBT) and EBRT respectively. However, isolated pelvic failure with either treatment is only two -3%. Any pelvic failure rate was 10% and 7% with VBT and EBRT respectively. Thus, most patients relapsed both in and outside pelvis or outside pelvis. One may choose VBT to avoid radiation toxicity, for EBRT may improve pelvic control but may only marginally improve survival if at all.
I think this article will become more interesting/useful if authors can find similar data/observations from the references cited and include in the discussion in this article. The readers will start thinking about ways to deal with LVSI. The manuscript is still good even If this were not possible. Minor revision refers only to this aspect.
Author Response
Answer: Thank you for this comment. The heterogeneity of adjuvant treatment for LVSI positive disease was the most striking observation in this analysis. We tried to ameliorate the clarity of the discussion on this topic by adding following paragraph: “This is particularly striking for nodal negative but LVSI positive endometrial cancer. For this situation there was no consensus among the experts, especially for pT1 disease. While some recommended BT alone, some preferred EBRT, and others a combination of EBRT plus BT. There were also some experts who did not recommend adjuvant treatment. However, the higher the grading the more experts recommended adjuvant treatment. At this, substantial LVSI and poorly differentiated histology are two of the most relevant risk factors for locoregional recurrence (32). The most recent ESGO/ESTRO/ESP guideline suggested adjuvant EBRT to be considered for stage I EC with substantial LVSI (especially those patients with G3 grading or omitted surgical lymph node staging) (6). PORTEC I and II showed that patients with substantial LVSI who received no adjuvant treatment or vaginal BT had a 5-year risk for pelvic regional recurrence of 25–30%, which was reduced to 5% by the addition of EBRT. It must be noted that only 5% of the patient cohorts in those two studies had substantial LVSI. As we have no significant survival difference in this specific cohort between BT or EBRT, and studies of accordingly selected patients for this entity are missing, the choice of treatment is often based on expert opinions as illustrated in this analysis. One may choose BT to avoid radiation toxicity, others may decide for EBRT to improve pelvic control but with non-significant survival benefit (33).”
Round 2
Reviewer 2 Report
This manuscript is revised version in the Decision-making analysis for Role of Brachytherapy in the postoperative management of endometrial cancer. Revision procedure performed well written but there is still the possibility of misleading future clinical practice. So, I do not think this manuscript is worth publishing in Cancers.
The decision of postoperative treatment for endometrial cancer should be made comprehensively by not only the radiation oncologist but also the gynecologist and medical oncologist.
As the role of chemotherapy in the adjuvant setting is still controversial, a consensus needs to be formed with mention of chemotherapy. It is considered that the consensus making without mention of chemotherapy may be misleading to future clinical practice. Author should be considered in this regard.
It is clear that there are large differences depending on the skill of the surgeon, even for the same R0 resection. This may not be a problem in facilities where surgeons are skilled enough (such as the ones who participated in this study), but it may lead to misleading clinical practice in facilities where surgeons are not skilled enough. Authors should indicate to readers that they need to have a good understanding of what surgical procedures are being performed at their institution in order to choose radiotherapy.
Author Response
We would like to thank the reviewer for this comment.
All experts participating in this study were suggested by the GEC-ESTRO gynaecological network group and have substantial experience in the field of gynaecological brachytherapy both within the framework of their respective national radiotherapy societies as well as active members of GEC-ESTRO working groups. They are all working in a leading position in high-volume gynecological cancer for brachytherapy. In all those centres, teatment decisions and recommendations are based on interdisciplinary tumour conferences which take full account of any input from the gynecological as well as medical oncology point of view. Therefore, the input from the experts analyzed in this work reflects their input within the framework of interdisciplinary tumour conferences. It is axiomatic that recommendations concerning the implementation of systemic treatments are subject of an interdisciplinary case evaluation, thus including also non-radiooncology expertise. To which extent, however, the indication of chemotherapy can be addressed in dependence from the surgical technique/procedure performed, cannot be satisfactorily evaluated since no uniform evidence, i.e. national/international guidelines, does so far clarify this point unambiguously.
In order to take due account of the constructive comment we added the following remark in the Discussion section: „As the role of chemotherapy in the adjuvant setting remains controversial with no uniform approach concerning its implementation, we plan to focus on this topic through a separate decision-making project which will not cover the role of specific radiotherapy modalities. Adding this question to the recent project would have make the analysis too complex given the intricacy of the underlying question. Notwithstanding, the aspect of adjuvant chemotherapy represents an ongoing topic and its indication should be evaluated within the framework of interdisciplinary tumour conferences which take full account of input from the gynecological as well as medical oncology point of view. The indication of adjuvant radiotherapy must likewise be evaluated accordingly.“
Reviewer 3 Report
Please describe why chemotherapy is excluded from the decision making tree in the Discussion.
Author Response
We would like to thank the reviewer for this comment.
As the role of chemotherapy in the adjuvant setting remains controversial with no uniform approach concerning its implementation, we decided, in accordance with the understanding of all participating experts, to focus on this topic through a separate decision-making project which will not cover the role of specific radiotherapy modalities. Adding this question to the recent project would make the analysis too complex given the intricacy of the underlying question. Therefore, we decided after careful consideration to focus only on radiotherapy approaches.
In order to take due account of the constructive comment we added the following remark in the Discussion section: „As the role of chemotherapy in the adjuvant setting remains controversial with no uniform approach concerning its implementation, we plan to focus on this topic through a separate decision-making project which will not cover the role of specific radiotherapy modalities. Adding this question to the recent project would have make the analysis too complex given the intricacy of the underlying question.“